# A High Protein Diet Is Associated with Improved Glycemic Control Following Exercise among Adolescents with Type 1 Diabetes

**DOI:** 10.3390/nu15081981

**Published:** 2023-04-20

**Authors:** Franklin R. Muntis, Abbie E. Smith-Ryan, Jamie Crandell, Kelly R. Evenson, David M. Maahs, Michael Seid, Saame R. Shaikh, Elizabeth J. Mayer-Davis

**Affiliations:** 1Department of Nutrition, Gillings School of Global Public Health, University of North Carolina, Chapel Hill, NC 27599, USA; frmuntis@email.unc.edu (F.R.M.);; 2Department of Exercise & Sports Science, University of North Carolina, Chapel Hill, NC 27519, USA; 3Department of Biostatistics, Gillings School of Global Public Health, University of North Carolina, Chapel Hill, NC 27599, USA; 4Department of Epidemiology, Gillings School of Global Public Health, University of North Carolina, Chapel Hill, NC 27599, USA; 5Division of Endocrinology, Department of Pediatrics, School of Medicine, Stanford University, Stanford, CA 94305, USA; 6Stanford Diabetes Research Center, Stanford, CA 94304, USA; 7Division of Pulmonary Medicine, Department of Pediatrics, Cincinnati Children’s Hospital, Cincinnati, OH 45229, USA; 8Department of Medicine, University of North Carolina, Chapel Hill, NC 27514, USA

**Keywords:** sports nutrition, type 1 diabetes, exercise, physical activity, glycemia, time-in-range, time-above-range, adolescents

## Abstract

Nutritional strategies are needed to aid people with type 1 diabetes (T1D) in managing glycemia following exercise. Secondary analyses were conducted from a randomized trial of an adaptive behavioral intervention to assess the relationship between post-exercise and daily protein (g/kg) intake on glycemia following moderate-to-vigorous physical activity (MVPA) among adolescents with T1D. Adolescents (*n* = 112) with T1D, 14.5 (13.8, 15.7) years of age, and 36.6% overweight or obese, provided measures of glycemia using continuous glucose monitoring (percent time above range [TAR, >180 mg/dL], time-in-range [TIR, 70–180 mg/dL], time-below-range [TBR, <70 mg/dL]), self-reported physical activity (previous day physical activity recalls), and 24 h dietary recall data at baseline and 6 months post-intervention. Mixed effects regression models adjusted for design (randomization assignment, study site), demographic, clinical, anthropometric, dietary, physical activity, and timing covariates estimated the association between post-exercise and daily protein intake on TAR, TIR, and TBR from the cessation of MVPA bouts until the following morning. Daily protein intakes of ≥1.2 g/kg/day were associated with 6.9% (*p* = 0.03) greater TIR and −8.0% (*p* = 0.02) less TAR following exercise, however, no association was observed between post-exercise protein intake and post-exercise glycemia. Following current sports nutrition guidelines for daily protein intake may promote improved glycemia following exercise among adolescents with T1D.

## 1. Introduction

Type 1 diabetes (T1D) is one of the leading causes of chronic disease in youth, with an estimated prevalence of 9 million people globally [1]. In 2017, the estimated prevalence of T1D among youth in the United States was 2.15 per 1000 youth, which represents a relative increase of 45.1% since 2001 [2]. Type 1 diabetes is associated with numerous health complications, including a risk of cardiovascular disease approximately ten times that of those without diabetes [3]. In the Diabetes Complications and Control Trial/Epidemiology of Diabetes Interventions and Complications Study (DCCT/EDIC), the extensive health benefits of intensive insulin therapy were highlighted, reporting a significant 57% and 42% reduction in cardiovascular disease events and mortality, respectively, among people with T1D. This same study, however, also observed that intensive insulin therapy was associated with weight gain and, among those classified as excessive weight gainers, the benefits of intensive insulin therapy were substantially diminished, with no difference in cardiovascular disease risk or mortality between those on intensive insulin therapy who gained excessive weight and those on conventional therapy at 6 years follow-up [4,5].

Participation in regular physical activity is a central part of both diabetes and weight management for adolescents with T1D. The American Diabetes Association recommends that adolescents with T1D participate in at least 60 min/day of moderate-to-vigorous physical activity (MVPA) [6]. Systematic reviews of physical activity and exercise interventions among youth with T1D indicate that regular physical activity is associated with improved glycemia, cardiorespiratory fitness, metabolic health, and weight management [7,8,9]. Despite these benefits of regular physical activity, research has shown that adolescents with T1D engage in lower levels of physical activity compared to their peers without diabetes, with as few as 37.8% achieving the World Health Organization (WHO) recommendations of at least 60 min of MVPA per day [10,11,12]. A major barrier to physical activity among adolescents with T1D is fear associated with the risk of experiencing hypoglycemia during and up to 24 h following exercise [13,14,15]. Of particular concern is an increased risk of hypoglycemia overnight, which may often result in more severe episodes of hypoglycemia or diabetic ketoacidosis [16,17,18]. Dietary guidelines, particularly nutrient timing recommendations, are needed to help guide safe participation in physical activity for adolescents with T1D.

While expert recommendations for carbohydrate intake before or after exercise for people with T1D have been published [19], less is known regarding the effects of protein intake on glycemia following physical activity. Sports nutrition guidelines recommend consumption of 0.25–0.3 g/kg or an absolute dose of 20–40 g of protein following exercise, as well as the consumption of high protein meals every 3–4 h following exercise to support recovery from and adaptation to an exercise bout [20,21]. Furthermore, in active individuals, a higher protein diet (25–30% energy from protein) combined with regular exercise has been associated with improved muscular strength and reduced soreness, and a significant decrease in fat mass when paired with a caloric deficit [22,23,24]. It is possible that similar protein recommendations may also improve glycemia following exercise for those with T1D, although minimal data exist in this area.

Among adolescents with T1D, protein intake has been associated with a mild hyperglycemic effect which persists for at least 5 h post-prandial [25,26,27], with one study suggesting that this effect may persist as long as 12 h following larger meals [28]. Only two studies, to the authors’ knowledge, have investigated the effects of protein intake on glycemia during or following exercise. One randomized controlled trial compared the effects of three different dietary approaches on glycemia during moderate-intensity cycling exercise among adolescents with T1D (*n* = 10): (1) a high protein breakfast (consumed two hours prior to exercise) plus a non-caloric placebo beverage (consumed 15 min prior to exercise), (2) a standard breakfast plus a carbohydrate beverage, (3) a standard breakfast plus a non-caloric placebo beverage [29]. The authors demonstrated that, while the carbohydrate beverage approach showed the slowest decline in glycemia during exercise, the protein-supplemented breakfast was equally effective at preventing hypoglycemia [29]. Additionally, a recent laboratory-based pilot study found that among 6 participants with T1D with a mean age of 20.2 ± 3.1 years, a 50 g protein whey protein bolus compared to water provided 3.25 h after 45 min of moderate exercise significantly reduced the glucose required to maintain euglycemia overnight [30]. While these studies support the theory that peri-exercise protein intake may reduce the risk of exercise-related hypoglycemia, the samples for these studies were small, and the highly controlled nature of the study designs may limit understanding of the efficacy of this nutritional strategy in a free-living environment.

As such, the primary aims of this study were to conduct secondary data analysis using data from a randomized trial of an adaptive behavioral intervention among adolescents with T1D to investigate the role of post-exercise (Aim 1) and daily protein intake (Aim 2) on glycemia following bouts of MVPA until the following morning. It was hypothesized that both post-exercise and daily protein intake would be associated with improvements in TIR and reductions in TBR following exercise among adolescents with T1D.

## 2. Materials and Methods

### 2.1. Study Design

To assess the proposed aims, post hoc analyses were performed using data from a randomized controlled trial of an adaptive behavioral intervention among adolescents with T1D named the Flexible Lifestyles Empowering Change (FLEX) study (1UC4DK101132-01). The FLEX study was conducted in accordance with the Declaration of Helsinki and was reviewed and approved by institutional review boards at clinical sites in Colorado and Ohio, as well as the coordinating site at the University of North Carolina at Chapel Hill (IRB #13-2856, Approved 10 March 2013). The FLEX study enrolled 258 adolescents with T1D between the ages of 13 and 16 years from 5 January 2014 to 4 April 2016. These participants were randomized to receive either usual care (*n* = 128) or an 18-month adaptive behavioral intervention (*n* = 130) aimed at improving diabetes self-management skills. The intervention utilized motivational interviewing and problem-solving skills training to help participants identify strategies to improve glucose control. While the intervention incorporated behavioral strategies around self-management skills, including insulin dosing, blood glucose testing, diet, and physical activity, the intervention did not systematically incorporate guidance for increasing physical activity. Written assent was provided by study participants, and written informed consent was provided by the study participants’ parents. These post hoc analyses utilize secondary measures from baseline and 6 months post-baseline visits to evaluate the proposed aims. Full details of the design and main results of the FLEX study have been published elsewhere [31,32].

### 2.2. Participants

The FLEX study recruited participants from two clinical sites: the Barbara Davis Center for Childhood Diabetes in Colorado and Cincinnati Children’s Hospital Medical Center in Ohio, with the University of North Carolina at Chapel Hill serving as a coordinating center from 5 January 2014 to 4 April 2016. Eligible criteria for the study included being between the ages of 13 and 16 years of age at study entry with a hemoglobin A1c (HbA1c) of 8–13% and diabetes duration of greater than one year. Youth who were pregnant or had severe concurrent physical, developmental, or psychiatric medical conditions were excluded from participating in the study. For the secondary analyses reported in this study, participants were included if they had reported a bout of MVPA at baseline or 6 months post-baseline visit and had sufficient dietary and glycemia data on the same day as the reported physical activity. Baseline demographic, clinical, glycemia, dietary, and physical characteristics among participants included in our analyses (*n* = 114) were evaluated and are reported in Table 1. Continuous variables are reported as mean and standard deviation except for non-normally distributed variables, in which median and interquartile range were reported. Categorical variables are described with counts and percentages.

### 2.3. Demographics and Health History

Demographic questionnaires were completed at baseline, from which self-reported age, sex, and race/ethnicity were reported. Health history questionnaires were completed at baseline, from which participants reported their date of diabetes diagnosis, insulin regimen, and total previous day insulin dose, among other measures. Follow-up health history questionnaires were administered 6 and 18 months post-intervention to report any changes in health history or diabetes care since their baseline visit. Age, sex, race/ethnicity, diabetes duration, insulin regimen, previous day insulin dose, and previous day insulin dose per kilogram of body weight were considered as potential covariates in our statistical models.

### 2.4. Continuous Glucose Monitoring (CGM)

Participants in the FLEX study were asked to wear a blinded Medtronic iPro2 continuous glucose monitor with an Enlite sensor for 7 days at baseline, 6 months, and 18 months post-intervention. As dietary and physical activity data were collected at baseline and 6 months, but not 18 months, CGM data from the 18-month visit are not included in these analyses. To enhance compliance and improve the quality of CGM data collection, an iPro2 compatible meter (OneTouch Ultra2) was provided to the participant along with 50 test strips for calibration 1 and 3 h after insertion, pre-meal, and before bed. Utilizing consensus report definitions [33], our outcomes of percent time in range (TIR, 70–180 mg/dL), percent time above range (TAR, >180 mg/dL), and percent time below range (TBR, <70 mg/dL) were calculated from the cessation of a bout of MVPA until 6:30 am the following morning to prevent confounding by dietary intake the following day. As the hyperglycemic effect of protein is known to last at least 5 h, observations with fewer than 5 h of CGM data following activity were excluded from our analyses.

### 2.5. Dietary Measures

During the 7-day CGM wear time, two unannounced 24 h dietary recalls were collected at baseline and 6 months post-intervention by certified interviewers from the UNC NIH/NIDDK Nutrition Obesity Research Center (NORC) staff (P30DK056350, MPI Mayer-Davis, Shaikh), using the Nutrient Data System for Research software and the multiple pass interviewing method [34,35]. For these analyses, participants with relative daily protein intake greater than three standard deviations above the mean (>3.21 g/kg) were excluded as potential outliers. Our Aim 1 exposure of post-exercise protein intake was defined as protein intake consumed between the end of a bout of MVPA and the end of the day (midnight) in both grams and grams/kg of body weight. Furthermore, as sports nutrition guidelines recommend daily protein intakes of 1.2–2.0 g/kg body weight to promote positive physiological adaptation to exercise [20], we further explored the effect of daily protein intake levels on glycemia for our Aim 2 analyses by comparing CGM metrics of TIR, TAR, and TBR from the cessation of MVPA bouts until the following morning between those who consumed <1.2 g/kg body weight and those who consumed ≥1.2 g per kilogram bodyweight.

### 2.6. Physical Activity Measures

At baseline and 6 months following the baseline visit, two previous day physical activity records (PDPAR) were collected during the 7-day CGM wear time by certified interviewers in conjunction with the 24 h dietary recalls. The PDPAR is a validated questionnaire that asks participants to describe the dominant activity and approximate intensity of activities they performed during the previous day in 30 min time blocks [36]. Intensities are described in categories as very light (slow breathing with little or no movement), light (normal breathing with regular movement), medium (increased breathing and quick movement for short periods of time), or hard (hard breathing with quick movement for ≥20 min). Each activity and perception of effort are matched to a corresponding metabolic equivalent (MET) value [36,37]. From these records, bouts of MVPA were defined as 30 min or greater of physical activity at a MET of greater than or equal to 3 METs. Average intensity (METs), bout duration (minutes), and bout volume (MET-minutes) were considered potential covariates in our statistical models.

### 2.7. Anthropometrics and Body Composition

Height, weight, and natural waist circumference were measured utilizing a wall-mounted stadiometer, calibrated electric scale, and a flexible fiberglass or steel tape measure, respectively, at baseline, 6, and 18 months post-intervention. From these measures, body fat percentage was estimated using validated age, race, and gender-specific equations [38]. Estimated body fat percentage was considered a potential covariate in our statistical models.

### 2.8. Statistical Analysis

#### 2.8.1. Model Selection

All statistical analyses were performed using SAS 9.4 (Cary, NC, USA). To account for repeated measures, mixed effect regression models were utilized for both our Aim 1 and Aim 2 analyses utilizing the Proc Mixed command. Potential covariates were introduced into our models in groups of design (study site, intervention group), demographic (age, sex, race/ethnicity), clinical (diabetes duration, insulin regimen, total previous day insulin dose, previous day insulin dose per kilogram body weight), body composition (estimated body fat percentage), physical activity (average bout intensity (METs), bout duration (minutes), bout volume (MET-minutes), other daily physical activity (MET-minutes)), dietary (daily carbohydrate intake, pre-exercise protein intake), and timing variables (hours until midnight). Covariates that caused a ≥10% change in the point estimate or standard error of associations were included in our final models.

#### 2.8.2. Aim 1 Analyses—Post-Exercise Protein Intake and Glycemia Following MVPA

Figure 1 provides an illustrated example timeline of exposures and outcomes relative to a bout of MVPA for both our Aim 1 and Aim 2 analyses. Post-exercise protein intake was defined continuously as protein intake (grams and grams/kg) from the cessation of a bout of MVPA until midnight. Mixed effects regression models assessed the association between post-exercise protein intake and TIR, TBR, and TAR from the cessation of a bout of MVPA until 6:30 am the following morning. Final models adjusted for the intervention group, study site, age, sex, race/ethnicity, diabetes duration, insulin regimen, estimated body fat percentage, MVPA bout volume (MET-minutes), other daily MVPA (MET-mins), daily carbohydrate intake, protein intake consumed within 4 h prior to exercise, and hours until midnight.

#### 2.8.3. Aim 2 Analyses—Overall Daily Protein Intake and Glycemia Following MVPA

To align with sports nutrition guidelines which recommend intakes of 1.2–2.0 g/kg/day of dietary protein to support exercise training, observations were categorized by daily protein intake into those with <1.2 g/kg/day of protein and those with ≥1.2 g/kg/day of protein utilizing a binomial variable. The relationship between protein intake category and CGM metrics of TIR, TBR, and TAR were assessed, with observations classified as <1.2 g/kg chosen as the reference group. The final analytic model adjusted for the intervention group, study site, age, sex, race/ethnicity, diabetes duration, insulin regimen, estimated body fat percentage, bout volume (MET-minutes), other daily MVPA (MET-mins), daily carbohydrate intake, and hours until midnight.

### 2.9. Exploration of Interaction Effects

Interaction effects were explored by sex, weight status, insulin regimen, MVPA bout volume (MET-mins), and whether a bout was vigorous (average bout MET-value ≥ 6.0) or moderate (average bout MET-value <6.0). The decision to include these terms was based on previous studies which have indicated that, among adolescents with T1D, those who utilize multiple daily insulin injections, those who have overweight or obesity, those with higher physical activity loads, and also female adolescents may experience more difficulties in managing glycemia which may then influence the association of protein intake on post-exercise glycemia in a free-living environment [39,40,41,42,43,44]. Interaction terms were added to the final Aim 1 and Aim 2 mixed effects regression models to assess for potential differences in response to post-exercise protein intake or daily protein intake category on glycemic metrics from the end of a MVPA bout until the following morning. Tanner score was also added as a covariate in statistical models assessing for differences by sex. Weight status was defined using BMI z-score to categorize participants by whether they had overweight/obesity or not at the time of their most recent study visit. Statistical significance for interaction effects was determined at a *p*-value <0.10.

## 3. Results

### 3.1. Final Sample Size

Of the 258 participants in the FLEX study, 135 participants reported at least 1 MVPA bout, with a total of 645 MVPA bouts identified. From these 162 bouts reported, 56 participants had insufficient or missing CGM data. Additionally, 7 bouts reported from 1 participant were excluded for reporting protein intakes above 3 standard deviations about the mean (>3.21 g/kg/day). Furthermore, 11 bouts (*n* = 5) were excluded for missing data on weight, 10 bouts (*n* = 1) were excluded for missing insulin regimen data, and 1 bout (*n* = 1) was excluded for missing sufficient data to estimate body fat percentage. As participants may have reported multiple bouts at both baseline and 6-month study visits, exclusion of a bout does not necessarily indicate full exclusion of a participant. Our final analytic models included 454 bouts from 114 participants, as detailed in Figure 2. In sensitivity analyses, we explored differences between FLEX participants included in our analyses and those not included (*n* = 114 vs. *n* = 144) in regard to the baseline characteristics provided in Table 1, and no significant differences were observed. Additionally, in the exploration of potential differences between bouts that were included versus those excluded for insufficient or missing CGM data (bouts = 454 vs. bouts = 162), no significant differences were observed for post-exercise or daily protein, carbohydrate, fat or calorie intake, any demographic variables included in Table 1, or any other variable included in our analytic models.

### 3.2. Baseline Characteristics

The baseline characteristics of FLEX participants included in our analyses are provided in Table 1. The median age, diabetes duration, and estimated body fat percentage for participants at baseline were 14.5 (IQR: 13.8, 15.7), 5.4 (IQR: 3.1, 9.0), and 28.1% (20.1%, 33.1%), respectively. There was similar inclusion of male (46.0%) and female (54.0%) participants. Most participants reported meeting WHO guidelines of achieving at least 60 min MVPA per day (97.3%). Furthermore, while the majority of participants (72.3%) reported using insulin pumps in their diabetes care, a majority of participants (69.9%) reported not using a continuous glucose monitor in their diabetes care in the past 30 days. Additionally, at baseline, participants had a median HbA1c of 9.3% (8.6%, 9.9%) and spent 36.4% ± 13.7% TIR, 59.7% ± 16.0% TAR, and 2.1% (IQR: 0.3%, 5.6%) TBR during their 7-day CGM wear time.

### 3.3. Aim 1 Results

The median bout duration (minutes) and intensity (METs) were 60 (IQR: 30, 90) and 4.5 (IQR: 4.0, 7.0), respectively. The average time from the cessation of MVPA bouts until midnight and 6:30 am the following morning was 9.0 ± 3.9 and 15.5 ± 10.4 h, respectively. The median protein intake from MVPA bout cessation until midnight was 34.9 (IQR: 20.9, 52.7) grams or 5.6 (IQR: 31.1, 0.86) grams/kilogram of body weight. The median TIR, TAR, and TBR from MVPA bout cessation until the following morning were 40.6% (IQR: 21.6%, 59.7%), 56.4% (IQR: 35.4%, 75.7%), and 0.00% (0.00%, 3.2%), respectively.

We observed no association between post-exercise protein intake and TAR, TIR, or TBR when examined in grams or grams per kilogram (Table 2). Additionally, we observed no statistically significant interaction effects between post-exercise protein intake and MVPA bout volume (*p*-values > 0.36), insulin regimen (*p*-values > 0.24), or weight status (*p*-values > 0.27) for TAR, TIR, or TBR. We did, however, observe a significant interaction between post-exercise protein intake in grams per kilogram (interaction *p* = 0.03) but not grams (interaction *p* = 0.16) with sex for TBR, indicating a significant association of −1.4% (95% CI: −1.7%, 0.0%, *p* = 0.05) TBR per 0.25 g/kg protein among female participants, but not male participants, −0.1% (95% CI: −0.5%, 0.7%, *p* = 0.76). Additionally, we observed significant interaction effects between post-exercise protein intake and sex when examined in grams (interaction *p* = 0.02) and grams per kilogram (interaction *p* = 0.03) with TIR of 3.6% (95% CI: 0.4%, 6.8%, *p* = 0.03) per 20 g and 2.3% (95% CI: −0.1%, 4.7%, *p* = 0.06) per 0.25 g/kg among females, but no association among males, −0.4% (−2.3%, 1.4%, *p* = 0.63) per 0.25 g/kg and 0.3% (95% CI: −2.7%, 2.0%. *p* = 0.78) per 20 g post-exercise protein. We did not observe a significant interaction effect between post-exercise protein intake in grams (interaction *p* = 0.10) or grams per kilogram (*p* = 0.19) with sex for TAR. Differences in the association between post-exercise protein and glycemia following exercise are depicted in Figure 3.

### 3.4. Aim 2 Results

The median daily protein intake reported on days with included bouts of MVPA was 65.4 (IQR: 48.2, 87.9) grams or 1.07 (IQR: 0.76, 1.48) grams/kg. Of the 454 bouts included in our final analyses, 188 bouts had daily protein intakes >1.2 g/kg/day, and 266 reported daily protein intakes below 1.2 g/kg/day. Aim 2 results are provided in Table 3. Daily protein intakes of ≥1.2 g/kg/day were associated with 8.0% (95% CI: 1.6%, 14.5%) less TAR and 6.9% (95% CI: 0.9%, 13.0%) greater TIR with no significant difference in TBR, 1.2% (95% CI: −0.8%, 3.2%). Additionally, we observed no significant interaction effect between the daily protein intake category and MVPA bout volume (*p*-values > 0.56) or whether a bout was vigorous or moderate (*p*-values > 0.29) for TAR, TIR, and TBR.

A significant interaction effect was observed between the daily protein intake category and insulin regimen for TIR (*p* = 0.03) and TAR (*p* = 0.08) but not for TBR, which indicated that adolescents who use multiple daily insulin injections (MDII) for their diabetes management may experience greater improvements in TIR, 17.9% (95% CI: 6.1%, 29.7%) and TAR, −17.9% (95% CI: −30.5%, −5.3%) (Table 4). Additionally, significant interaction effects were observed between protein intake category and weight status for TIR (*p* < 0.01), TBR (*p* < 0.01), and TAR (*p* = 0.08), indicating that following a high protein diet pattern may improve TIR, 18.6% (95% CI: 8.7%, 28.4%), and TAR,−15.6% (95% CI: −26.1%, −5.1%) to a greater extent for adolescents with overweight/obesity, but may also elevate TBR, 2.7% (95% CI: 0.6%, 4.9%) among normal weight individuals. Following additional adjustment for the tanner stage, significant interaction effects were also observed between protein intake category and sex for TIR (*p* < 0.01) and TAR (*p* < 0.01), indicating that following a high protein diet pattern may improve TIR, 16.3% (95% CI: 8.4%, 24.2%), and TAR, −16.9% (−25.3%, −8.5%), to a greater extent among female adolescents. Figure 4 shows a comparison of the effects of a high protein diet on post-exercise glycemia by insulin regimen, weight status, and sex.

## 4. Discussion

This study evaluated a unique intersection between nutrient timing and diabetes care by assessing the effects of post-exercise and daily protein intakes on post-exercise glycemia among adolescents with T1D. It was hypothesized that increased post-exercise and daily protein intake would be associated with improved TIR and reduced TBR following cessation of MVPA bouts until the following morning. No significant associations were observed between post-exercise protein intake (g or g/kg) and any primary outcome. Further, no significant interaction effects were observed between post-exercise protein intake and exercise volume, insulin regimen, or weight status; however, a significant interaction effect was observed for post-exercise protein intake and sex, which indicated that increased post-exercise protein intake may be associated with improved TIR and reduce TBR among female, but not male adolescents. Additionally, daily protein intakes ≥1.2 g/kg/day were associated with improved TIR and reduced TAR, but not TBR following MVPA bouts, with significant interaction effects observed indicating that female adolescents, those with overweight/obesity, and those on multiple daily insulin injections may experience greater increases in TIR and greater reductions in TAR with daily protein intakes >1.2 g/kg/day.

The findings of this study did not support the hypothesis that increasing post-exercise protein intake may improve glycemia following exercise among adolescents with T1D overall, but it may be a beneficial strategy among female adolescents with T1D. The timing of MVPA bouts, however, may be important to consider, as most of the reported bouts of MVPA in this study occurred in the afternoon or evening. As afternoon exercise is associated with greater post-exercise hypoglycemia risk among people with T1D [15], it would be expected that if protein had a protective effect against hypoglycemia that it would be observed following afternoon exercise [15]; however, it is also important to consider that many adolescents may have their last meal of the day in the early evening, making the size of post-exercise protein exposure somewhat limited (median intake of 34.9 (IQR: 20.9, 52.7) g). The few studies which have examined the effects of protein on glycemia over a period of time similar to that assessed in this study utilized larger protein doses (≥60 g) [28,45]. It is possible that a larger protein dose may be necessary to promote changes in glycemia overnight. The findings of this study did support the hypothesis that elevated daily protein intakes, within sports nutrition, recommended daily intake levels of 1.2–2.0 g/kg/day, may improve the post-exercise glycemic response, especially among individuals utilizing multiple daily insulin injections for their diabetes care, those with overweight/obesity and among female adolescents.

While research on the effects of dietary protein intake on exercise-related glycemia among people with T1D is relatively scarce, a recent laboratory-based pilot study found that a protein bolus of 50 g following moderate-intensity exercise caused elevated glucagon, glucagon-like peptide-1 (GLP-1) and gastric inhibitory peptide (GIP) levels overnight compared to water following exercise, which collectively led to reduced glucose infusion requirements to maintain euglycemia [30]. Studies in people with type 2 diabetes (T2D) have observed similar increases in the gut hormone GLP-1 as well as a pancreatic polypeptide (PYY)with high protein diets, which have been shown to suppress the rate of gastric emptying and, therefore, the rate at which blood glucose concentrations increase following a meal [46,47]. Additionally, high protein diets have been shown to improve insulin sensitivity via reductions in intra-hepatic liver triglycerides and increases in post-meal glucagon secretion among people with T2D [48,49,50,51]. It is possible that improvements in insulin sensitivity and reductions in gastric emptying rate associated with elevated protein intakes may contribute to the improvements in post-exercise glycemia observed in this study.

These prior research findings may also help explain why people on multiple daily insulin injections, or those with overweight or obesity, experience greater improvements in post-exercise glycemia with elevated daily protein intakes. Individuals on multiple daily insulin injections have been shown to experience greater levels of post-exercise hyperglycemia compared to their peers who use insulin pumps which may be attributed in part to greater carbohydrate consumption to avoid hypoglycemia among individuals on multiple daily insulin injections who have less acute control over insulin dosing compared to insulin pump users [39]. As such, reductions in gastric emptying rate may slow the rise in glycemia following meals containing both carbohydrates and protein, which could contribute to less TAR and also more TIR. Additionally, elevated adiposity among individuals with overweight or obesity has been associated with higher levels of insulin resistance in youth [52]. In fact, among adolescents and adults with T1D, fat mass has been shown to be positively related to post-exercise blood glucose, and lean mass has been shown to be inversely related to post-exercise blood glucose, indicating that body composition may play an important role in the post-exercise glycemic response [53]. Changes in insulin sensitivity may have contributed to improved post-exercise glycemia to a greater extent among this population.

Differences in body composition between male and female participants may also help to explain the differences in the association we observed by sex, as female participants in this study had higher estimated body fat percentages compared to male participants (33.5% ± 5.9% vs. 20.3% ± 4.2%). While we adjusted our models for estimated body fat percentage, we may not fully account for other differences in body composition, including differences in lean mass between male and female adolescents. Additionally, previous studies in adolescents with T1D have shown that female adolescents oxidize more fat and less carbohydrate during exercise compared to males [54]. In populations without diabetes, this difference in substrate utilization among women has been attributed in part to differing progesterone and estrogen levels during the follicular versus luteal phase of the menstrual cycle and has been shown to yield less hepatic and muscular glycogen depletion during exercise among women [55]. While speculative, these differences in substrate utilization during exercise and the potential sparing of hepatic and muscle glycogen may influence the post-exercise glycemic response among female adolescents. Studies are needed, however, to elucidate the mechanisms by which protein affects glycemia and the influence of factors such as insulin regimen, weight status, and sex on this relationship in the unique metabolic context of T1D.

### 4.1. Significance for Clinical Practice

As most exercise nutrition studies among people with T1D to date have focused predominantly on carbohydrate or insulin dosing strategies to improve exercise-related glycemia, the current study addresses an important gap in the existing evidence and can inform exercise nutrition guidelines regarding the role of protein intake on exercise-related glycemia for people living with T1D. Sports nutrition guidelines currently recommend daily protein intakes of 1.2–2.0 g/kg/day as an effective strategy for improving recovery, athletic performance, and weight management when combined with exercise training [20,21]. These guidelines were largely based on healthy populations; however, it is likely that people with T1D may experience similar benefits following these protein intake recommendations. The findings of this study suggest higher protein intakes may also help adolescents with T1D improve their post-exercise glycemic response, especially among female adolescents and also among those who do not utilize insulin pumps in their diabetes care, and those with overweight or obesity. We did, however, observe that adolescents without overweight or obesity may experience higher TBR following a higher protein diet. As such, additional counseling and monitoring may be needed to support adolescents with T1D who may choose to follow a higher protein diet to support athletic goals.

### 4.2. Challenges and Opportunities

It is important to note that data reported in this study are observational and future work is needed to establish whether a causal relationship exists between dietary protein intake and post-exercise glycemia among individuals with T1D. Additionally, as this study relied on self-reported measures of dietary intake and physical activity, it’s important to note that self-reported measures are prone to recall and social desirability biases [56,57]. Specifically, dietary intake has commonly been shown to be under-reported while MVPA is often over-reported when compared to accelerometry among adolescents which may influence the number of MVPA bouts identified in this study [56,57]. The use of the multiple pass method for dietary recalls, however, has been shown to reduce bias in self-reported dietary intake, and self-reported MVPA has been shown to be more reliably measured when collected by trained interviewers, as was done in the FLEX study [35,58,59]. Additionally, the PDPAR instrument utilized in this study has been validated against accelerometry for relative energy expenditure of physical activity (r = 0.77, *p* < 0.01) and has been shown to provide reliable identification of bouts of MVPA on a previous day (r = 0.63, *p* < 0.01) [36,60]. Additionally, the lack of time-stamped insulin-dosing data for these analyses limits our ability to understand the role of insulin-dosing behaviors on the observed associations. While controlling for daily carbohydrate intake may help to account for bolus insulin levels, as bolus insulin doses are based on carbohydrate intake, we cannot account for basal insulin dosing or potential insulin dosing strategies which may have been implemented to reduce the risk of exercise-related hypoglycemia. Additionally, we did not collect data related to hormone levels or menstrual cycle among participants and therefore are unable to elucidate the role these factors may play in the differing effects of protein intake on post-exercise glycemia observed in our study. However, the availability of time-stamped dietary intake, physical activity, and continuous glucose monitoring data from the FLEX study provided a unique opportunity to assess a temporal relationship between protein intake and post-exercise glycemia among adolescents with T1D.

The literature on the role of protein intake on exercise-related glycemia for people living with T1D is scarce. The findings of this study begin to address this gap in the literature and may encourage future studies to continue to explore intersections between sports nutrition and diabetes care. While promoting safe exercise through an improved glycemic response is a priority for people living with T1D, they also chose to participate in exercise for a variety of reasons, including health promotion, weight management, and athletic performance. It is important that exercise nutrition guidelines aim to support both the safety and physiologic benefits of exercise to aid people with T1D in improving their health and well-being.

### 4.3. Future Research Directions

Randomized controlled trials are needed to elucidate whether a causal relationship exists between dietary protein intake and exercise-related glycemia among people with T1D and to identify potential mechanisms of action for which protein may affect the post-exercise glycemic response. Additionally, the use of mixed methods research may provide invaluable insight into practical aspects of this nutritional strategy, such as perceptions of the feasibility and potential barriers to implementing this dietary approach among adolescents and adults living with T1D. Additionally, as the fear of hypoglycemia is a leading barrier to regular physical activity among people with T1D, future research should aim to further evaluate the effects of following a high protein diet on the risk of hypoglycemia among people with T1D, specifically following exercise and overnight when the risk of experiencing severe hypoglycemia is heightened [16,17,18]. Finally, while the benefits of elevated protein intake on the adaptive response to exercise have been well-documented in healthy populations [20,21], research is needed to evaluate whether these adaptive benefits are similar among people living with T1D.

## Figures and Tables

**Figure 1 nutrients-15-01981-f001:**
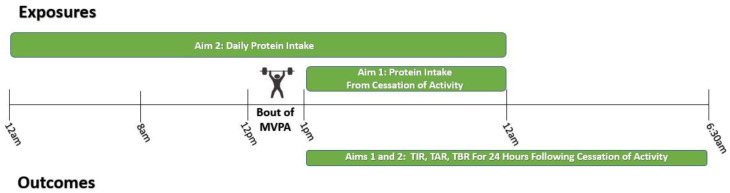
Example timeline of exposures and outcomes relative to a bout of moderate-to-vigorous physical activity using multiple measurements (baseline and 6 months). Individuals may report multiple bouts of moderate-to-vigorous physical activity (MVPA) per day.

**Figure 2 nutrients-15-01981-f002:**
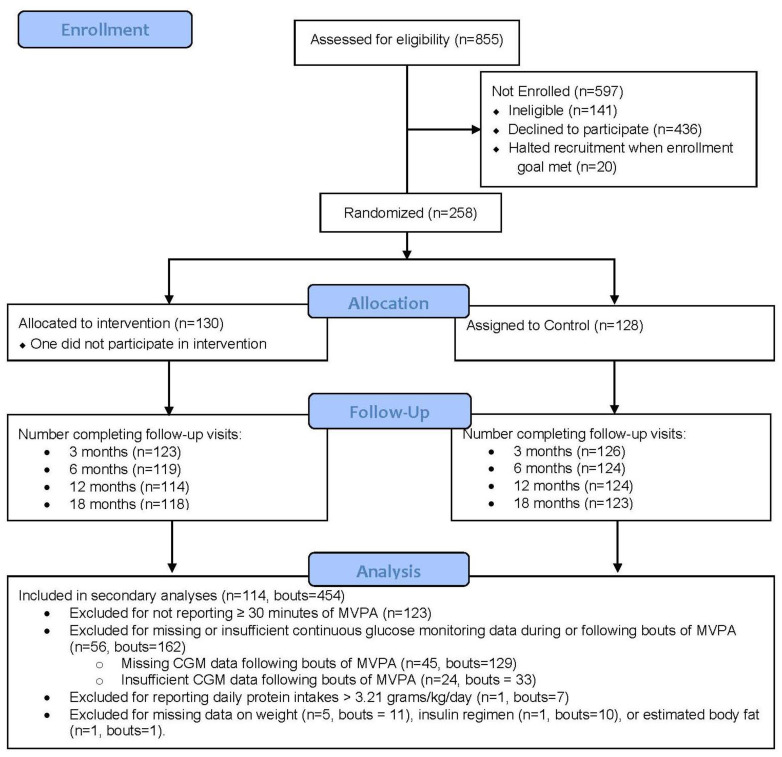
CONSORT flow diagram for secondary analyses of the flexible lifestyles empowering change (FLEX) randomized trial.

**Figure 3 nutrients-15-01981-f003:**
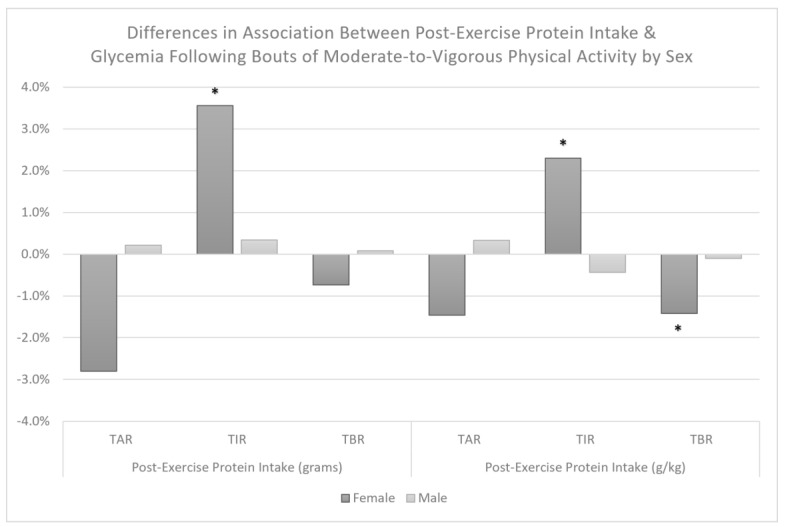
Differences in association between post-exercise protein intake and glycemia from cessation bouts of MVPA until the following morning by sex among adolescents with T1D. Estimates are provided per 20 g or 0.25 g/kg. TAR = percent time above range (>180 mg/dL), TIR = percent time in recommended glucose range (70–180 mg/dL), and TBR = percent time below range (<70 mg/dL) following bouts of MVPA. * Indicates an association that is statistically significant (*p* ≤ 0.05).

**Figure 4 nutrients-15-01981-f004:**
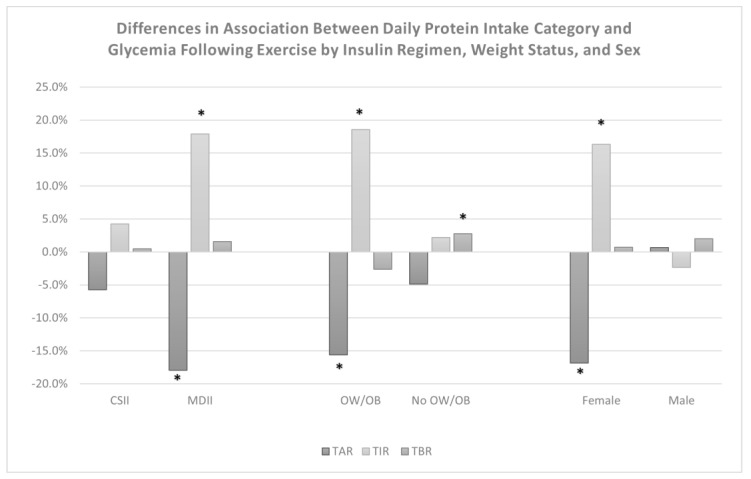
Differences in the effect of consuming a higher protein diet (>1.2 g/kg/day) on glycemia from the end of MVPA bouts until the following morning by insulin regimen, weight status, and sex. CSII = continuous subcutaneous insulin infusion, MDII = multiple daily insulin injections, OW/OB = overweight or obesity, No OW/OB = no overweight or obesity. TAR = percent time above range (>180 mg/dL), TIR = percent time in recommended glucose range (70–180 mg/dL), TBR = percent time below range (<70 mg/dL) following bouts of MVPA. * Indicates an association that is statistically significant (*p* ≤ 0.05).

**Table 1 nutrients-15-01981-t001:** Baseline characteristics of FLEX participants included in final analyses (*n* = 114).

Demographic	Mean ± SD or *n* (%)
Age	14.5 (13.8, 15.7)
Female	61 (54.0)
Male	52 (46.0)
Race/Ethnicity	
Non-Hispanic White	91 (80.5)
Non-Hispanic Black	2 (1.8)
Hispanic	14 (12.4)
Multiracial/Other	6 (5.3)
Maximum Education of Parents	
High School or Less	11 (9.8)
Some College	31 (27.7)
Four-Year College Degree	50 (44.6)
Graduate Degree	20 (17.9)
Clinical	
Diabetes Duration	5.4 (3.1, 9.0)
Insulin Pump User (*n* = 111)	81 (72.3)
Previous Day Insulin Dose (units/kg) (*n* = 110)	1.0 ± 0.3
Anthropometric	
Weight (kg)	58.8 (51.3, 69.2)
BMI Z-Score	0.7 ± 0.9
Estimated Body Fat %	28.1 (20.1, 33.1)
Glycemia	
No Personal CGM Use in Past 30 Days (*n* = 103)	72 (69.9)
Baseline HbA1c (%)	9.3 (8.6, 9.9)
Percent Time in Range (*n* = 106)	36.4 ± 13.7
Percent Time Below Range (*n* = 106)	2.1 (0.3, 5.6)
Percent Time Above Range (*n* = 106)	59.7 ± 16.0
Diet	
Daily Caloric Intake (kcal)	1623.3 (1315.6, 2062.0)
Percent of Daily Calories from Protein	16.0 ± 3.5
Percent of Daily Calories from Carbohydrates	49.0 ± 7.7
Percent of Daily Calories from Fat	36.2 ± 6.4
Daily Fiber Intake (grams)	13.4 (10.2, 18.2)
Physical Activity (*n* = 109)	
Meet WHO Guidelines of ≥60 min MVPA/day	101 (92.7)
Daily Minutes of MVPA	165.0 (105.0, 225.0)
Daily Minutes of Vigorous Physical Activity	45.0 (0.0, 90.0)

Continuous variables are reported as mean and standard deviation except for non-normally distributed variables, in which median and interquartile range are reported. Categorical variables are described with counts and percentages.

**Table 2 nutrients-15-01981-t002:** Results of mixed effects regression models assessing the association between post-exercise protein intake and glycemia following a bout of MVPA until 6:30 am the following morning among adolescents with type 1 diabetes (*n* = 114, bouts = 454).

	Post-Exercise Protein (Grams) *			Post-Exercise Protein (g/kg) †		
	Estimate	*p*-Value	95% CI	Estimate	*p*-Value	95% CI
	Unadjusted Models					
Percent Time Above Range	0.5%	0.52	(−1.1%, 2.2%)	0.6%	0.33	(−0.6%, 1.9%)
Percent Time In Range	−0.4%	0.58	(−2.0%, 1.1%)	−0.6%	0.35	(−1.7%, 0.6%)
Percent Time Below Range	0.1%	0.63	(−0.6%, 0.4%)	−0.1%	0.77	(−0.4%, 0.3%)
	Fully Adjusted Models ‡					
Percent Time Above Range	−0.7%	0.56	(−3.0%, 1.6%)	−0.1%	0.93	(−1.8%, 1.6%)
Percent Time In Range	0.8%	0.49	(−1.4%, 2.9%)	0.2%	0.31	(−1.4%, 1.8%)
Percent Time Below Range	0.1%	0.79	(−0.8%, 0.6%)	−0.1%	0.66	(−0.7%, 0.4%)

* Associations are reported per a 20 g dose of protein; † Associations are reported per a 0.25 g/kg dose of protein; ‡ Models are adjusted for design (study site, intervention group), demographic (age, sex, race/ethnicity), clinical (diabetes duration, insulin regimen), anthropometric (estimated body fat percentage), dietary (daily carbohydrate intake and pre-exercise protein intake), physical activity (bout volume, other daily physical activity), and hours until midnight.

**Table 3 nutrients-15-01981-t003:** Results of linear mixed-effects regression models comparing continuous glucose monitoring metrics following cessation of bouts of moderate-to-vigorous physical activity until 6:30 am the following morning by category of daily protein intake (g/kg/day) among adolescents with type 1 diabetes (*n* = 114, bouts = 454).

Category of Daily Protein Intake	% Time above Range			% Time in Range			%Time below Range		
	Estimate	*p*-Value	95% CI	Estimate	*p*-Value	95% CI	Estimate	*p*-Value	95% CI
	Unadjusted Models								
<1.2 g Protein/kg Body weight (bouts = 266)	Reference								
>1.2 g Protein/kg Body weight (bouts = 188)	−6.8%	0.02	(−12.4%, −1.1%)	5.3%	0.05	(0.0%, 10.6%)	1.5%	0.09	(−0.3%, 3.2%)
	Fully Adjusted Models *								
<1.2 g Protein/kg Body weight (bouts = 266)	Reference								
>1.2 g Protein/kg Body weight (bouts = 188)	−8.0%	0.02	(−14.5%, −1.6%)	6.9%	0.03	(0.9%, 13.0%)	1.2%	0.22	(−0.8%, 3.2%)

* Final models were adjusted for design (intervention group, study site), demographic (age, sex, race/ethnicity), clinical (diabetes duration, insulin regimen), anthropometric (estimated body fat percentage), physical activity (bout volume (MET-mins), other daily MVPA (MET-mins), dietary (daily carbohydrate intake), and timing (hours until midnight) variables.

**Table 4 nutrients-15-01981-t004:** Results of mixed-effects regression models assessing interaction between daily protein intake category with insulin regimen and weight status.

Interaction Effects *	% Time above Range			% Time in Range			% Time below Range		
	Estimate	*p*-Value	95% CI	Estimate	*p*-Value	95% CI	Estimate	*p*-Value	95% CI
Protein Intake Category × Insulin Regimen	Interaction *p*-Value = 0.08			Interaction *p*-Value = 0.03			Interaction *p*-Value = 0.60		
Continuous Subcutaneous Insulin Infusion (CSII)	−5.7%	0.1	(−12.5%, 1.1%)	4.2%	0.19	(−2.2%, 10.6%)	0.5%	0.81	(−3.4%, 4.4%)
Multiple Daily Insulin Injections (MDII)	−17.9%	<0.01	(−30.5%, −5.3%)	17.9%	<0.01	(6.1%, 29.7%)	1.6%	0.13	(−0.55%, 3.8%)
Protein Intake Category × Weight Status	Interaction *p*-Value = 0.08			Interaction *p*-Value <0.01			Interaction *p*-Value <0.01		
Overweight/Obesity	−15.6%	<0.01	(−26.2%, −5.1%)	18.6%	<0.001	(8.7%, 28.4%)	−2.6%	0.11	(−5.8%, 0.6%)
No Overweight/Obesity	−4.9%	0.18	(−12.0%, 2.3%)	2.2%	0.52	(−4.5%, 8.8%)	2.7%	0.01	(0.6%, 4.9%)
Protein Intake Category × Sex	Interaction *p*-Value <0.01			Interaction *p*-Value <0.01			Interaction *p*-Value =0.48		
Female	−16.9%	<0.0001	(−25.3%, −8.5%)	16.3%	<0.001	(8.4%, 24.2%)	0.7%	0.61	(−2.0%, 3.4%)
Male	0.6%	0.88	(−7.7%, 9.0%)	−2.4%	0.56	(−10.3%, 5.5%)	2.0%	0.14	(−0.7%, 4.7%)

* Interaction models estimated the combined effect of increasing protein intake category from <1.2 g/kg/day to ≥1.2 g/kg/day and category of insulin regimen (CSII/MDII) or weight status (has overweight/obesity or does not have overweight/obesity). Mixed-effects regression models were adjusted for design (intervention group, study site), demographic (age, sex, race/ethnicity), clinical (diabetes duration, insulin regimen), anthropometric (estimated bodyfat percentage (protein × Insulin regimen), or weight status (protein × weight status)), physical activity (bout volume (MET-mins), other daily MVPA (MET-mins)), dietary (daily carbohydrate intake), and timing (hours until midnight) variables. Interaction models for sex and category of protein intake were additionally adjusted for the tanner stage.

## Data Availability

The data in this study are openly available in the NIDDK Data Repository at doi: 10.58020/235v-4k70.

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
