# Peer review of "A High Protein Diet Is Associated with Improved Glycemic Control Following Exercise among Adolescents with Type 1 Diabetes"

_nutrients, 2023, doi:10.3390/nu15081981_

Round 1

Reviewer 1 Report

Diet management is very important in patients with either type of diabetes mellitus. Patients with type 1 diabetes must adjust their insulin dose in relation to carbohydrate intake or the carbohydrate exchange system to avoid hypoglycemia.  The physical effort can help patients with diabetes mellitus to control their weight and blood glucose levels.  Because blood glucose levels go down during physical effort, patients must be alert for symptoms of hypoglycemia. Some patients need to eat a small snack during prolonged exercise or to decrease their insulin dose or both. 

This study evaluated the importance of synchronizing nutrient intake with exercise and diabetes treatment and presented the effects of daily protein intake after exercise on post-exercise blood glucose in adolescents with type 1 diabetes.

Mixed effects regression models assessed the association between post-exercise protein intake and TIR, TBR and TAR after a moderate or great physical effort. 

High protein diets have been shown to improve insulin sensitivity via reductions in intrahepatic triglycerides and increases in post-meal glucagon secretion among people with diabet mellitus. It is possible that improvement in insulin sensitivity and reduction in gastric emptying rate associated with elevated protein intakes may contribute to the improvements in post-exercise glycemia. 

The study provides an important perspective on the practical aspects of this nutritional strategy and the potential barriers to this dietary approach among adolescents and young adults with type 1 diabetes. 

-

Author Response

Response to Reviewer’s Feedback
Reviewer 1’s Comments to Authors:
Diet management is very important in patients with either type of diabetes mellitus. Patients with
type 1 diabetes must adjust their insulin dose in relation to carbohydrate intake or the
carbohydrate exchange system to avoid hypoglycemia. The physical effort can help patients with
diabetes mellitus to control their weight and blood glucose levels. Because blood glucose levels
go down during physical effort, patients must be alert for symptoms of hypoglycemia. Some
patients need to eat a small snack during prolonged exercise or to decrease their insulin dose or
both.
This study evaluated the importance of synchronizing nutrient intake with exercise and diabetes
treatment and presented the effects of daily protein intake after exercise on post-exercise blood
glucose in adolescents with type 1 diabetes.
Mixed effects regression models assessed the association between post-exercise protein intake
and TIR, TBR and TAR after a moderate or great physical effort.
High protein diets have been shown to improve insulin sensitivity via reductions in intrahepatic
triglycerides and increases in post-meal glucagon secretion among people with diabetes mellitus.
It is possible that improvement in insulin sensitivity and reduction in gastric emptying rate
associated with elevated protein intakes may contribute to the improvements in post-exercise
glycemia.
The study provides an important perspective on the practical aspects of this nutritional strategy
and the potential barriers to this dietary approach among adolescents and young adults with type
1 diabetes.
Thank you for your positive feedback and comments on the importance of this work. We believe these
findings are a strong initial step in providing insight into the role of dietary protein intake on the
management of glycemia following exercise for people living with T1D. We hope that this work will
encourage further investigation into this important area of research and encourage others to further
bridge sports nutrition and diabetes care to identify strategies that can aid people with T1D in improving
their health, fitness and diabetes management.

Reviewer 2 Report

This is a well-written manuscript with the findings obtained through proper statistical methods. I have a few comments for the authors to consider. 

1. Will you be able to delineate protein from diet and protein from supplements? Similarly plant versus animal-based protein?

2. Abstract: Your findings and the conclusion seem to conflict with each other. Please check. 

3. What is the rationale for choosing certain participant characteristics to test for interactions?

4. Do you think the amount of protein intake recommended by the guidelines should be increased? If yes, is there any data to support this?

5. Are there any sex-specific differences in glycemic control following post-exercise protein intake, given the differences in muscle mass between males and females?

Author Response

Response to Reviewer’s Feedback
Reviewer 2’s Comments to Authors:
This is a well-written manuscript with the findings obtained through proper statistical methods. I
have a few comments for the authors to consider.
1. Will you be able to delineate protein from diet and protein from supplements? Similarly plant
versus animal-based protein?
Unfortunately, our data does not allow us to delineate protein from diet and protein from supplements,
however, as our population in this study was adolescents with suboptimal glycemic control, we expect
that protein supplement use was likely minimal among this group. We also appreciate your suggestion to
look at plant versus animal-based protein. While we agree that considering the potential differential
effects of animal versus plant protein is important to consider, this population included adolescents with
suboptimal glycemic control and the amount of daily plant-based protein intake was small with a median
daily intake of 23.3 grams (IQR: 14.9, 30.7). Therefore, we have insufficient variability in intake to
adequately assess the role of plant-based versus animal-based protein intake on post-exercise glycemia.
In future work that builds off the results of this study, however, we agree that it will be important to
differentiate between plant and animal-based proteins and we will be intentional about designing future
studies to better evaluate the role they each play in glycemia among adolescents with T1D.
2. Abstract: Your findings and the conclusion seem to conflict with each other. Please check.
We apologize for the confusion. In our Aim 1 analyses, which specifically looked at protein intake
consumed after exercise, we did not observe a significant association between post-exercise protein
intake and post-exercise glycemia. In our Aim 2, however, we noted that overall daily intakes within sports
nutrition guidelines of 1.2-2.0g/kg/day was associated with improved glycemia following exercise
suggesting a diet that meets sports nutrition guidelines for daily protein intake may be associated with
improved post-exercise glycemia. We adjusted the results section of our abstract, highlighting our main
findings first to improve the clarity of our findings as follows:
“Daily protein intakes of ≥1.2g/kg/day were associated with 6.9% (p=0.03) greater TIR and -8.0% (p=0.02)
less TAR following exercise, however, no association was observed between post-exercise protein intake
and post-exercise glycemia. Following current sports nutrition guidelines for daily protein intake may
promote improved glycemia following exercise among adolescents with T1D.”
3. What is the rationale for choosing certain participant characteristics to test for interactions?
Our decision to look at interactions by insulin regimen, weight status, and bout volume/intensity were
based off previous studies which have shown that adolescents who utilize multiple daily insulin injections
(MDII), those with overweight and obesity, and those with higher physical activity loads may have
worsened glycemic control in comparison to their peers who use continuous subcutaneous insulin
infusion (CSII), those without overweight/obesity, or those with lower physical activity loads and therefore
may have experience more or less benefit to their glycemic control in response to post-exercise or daily
protein intake. Additionally, we have added sex as an interaction term based on the feedback you shared.
We have added the following to the manuscript:
In the methods section on Page 6:
“Interaction effects were explored by sex, weight status, insulin regimen, MVPA bout volume (MET-mins),
and whether a bout was vigorous (average bout MET-value ≥ 6.0) or moderate (average bout MET-value
<6.0). The decision to include these terms was based off of previous studies which have indicated that,
among adolescents with T1D, those who utilize multiple daily insulin injections, those who have
overweight or obesity, those with higher physical activity loads and also female adolescents may
experience more difficulties in managing glycemia which may then influence the association of protein
intake on post-exercise glycemia in a free-living environment.39-44”
4. Do you think the amount of protein intake recommended by the guidelines should be
increased? If yes, is there any data to support this?
That is an excellent question. While there are expert consensus guidelines that provide guidance on
carbohydrate or insulin dosing strategies to support exercise for people with type 1 diabetes, there is little
guidance on protein intake recommendations to support exercise. We hope the findings of this study will
help to inform future exercise nutrition guidelines in this population and will encourage further research in
this area which may establish more concrete guidelines on protein dosage and timing to promote both
improved glycemia and adaptive benefits for people living with T1D, however, currently there isn’t
sufficient evidence to support recommendations to increase protein intake guidelines.
5. Are there any sex-specific differences in glycemic control following post-exercise protein
intake, given the differences in muscle mass between males and females?
Great question! We had originally considered looking at sex-related differences, but we were concerned
that due to our population being adolescents that sex-related differences may be confounded by
differences in pubertal stage among participants. Following your recommendation, however, we decided
to explore potential interactions by sex with the addition of tanner stage as a covariate in our statistical
models to help to account for potential differences by pubertal stage. In doing so, we did identify several
significant interaction effects by sex. We have added details about these observed interaction effects in
the text as follows:
In the Method’s Section on Page 6, under 2.9 Exploration of Interaction Effects:
“Interaction effects were explored by sex, weight status, insulin regimen, MVPA bout volume (MET-mins),
and whether a bout was vigorous (average bout MET-value ≥ 6.0) or moderate (average bout MET-value
<6.0). The decision to include these terms was based off of previous studies which have indicated that,
among adolescents with T1D, those who utilize multiple daily insulin injections, those who have
overweight or obesity, those with higher physical activity loads and also female adolescents may
experience more difficulties in managing glycemia which may then influence the association of protein
intake on post-exercise glycemia in a free-living environment.39-44 Interaction terms were added to the
final Aim 1 & Aim 2 mixed effects regression models to assess for potential differences in the response to
post-exercise protein intake or daily protein intake category on glycemic metrics from the end of a MVPA
bout until the following morning. Weight status was defined using BMI z-score to categorize participants
by whether they had overweight/obesity or not at the time of their most recent study visit. Statistical
significance for interaction effects were determined at a p-value <0.10.”
In Aim 1 Results on Page 9:
“We did, however, observe a significant interaction between post-exercise protein intake in grams per
kilogram (interaction p=0.03), but not grams (interaction p=0.16) with sex for TBR, indicating a significant
association of -1.4% (95% CI: -1.7%, 0.0%, p=0.05) TBR per 0.25g/kg protein among female participants,
but not male participants, -0.1% (95% CI: -0.5%, 0.7%, p=0.76). Additionally, we observed significant
interaction effects be-tween post-exercise protein intake and sex when examined in grams (interaction
p=0.02) and grams per kilogram (interaction p=0.03) with TIR of 3.6% (95% CI: 0.4%, 6.8%, p=0.03) per
20g and 2.3% (95% CI: -0.1%, 4.7%, p=0.06) per 0.25g/kg among females, but no association among
males, -0.4% (-2.3%, 1.4%, p=0.63) per 0.25g/kg and 0.3% (95% CI: -2.7%, 2.0%. p=0.78) per 20g postexercise protein. We did not observe a significant interaction effect between post-exercise protein intake
in grams (interaction p=0.10) or grams per kilogram (p=0.19) with sex for TAR. Differences in association
between post-exercise protein and glycemia following exercise are depicted in Figure 2.”
We have also added the following Figure 3 on Page 10 which provides an illustrated comparison of the
differences in the association between post-exercise protein intake and post-exercise glycemia by sex:
Figure 3. Differences in association between post-exercise protein intake and glycemia from cessation
bouts of MVPA until the following morning by sex among adolescents with T1D. Estimates are provided
per 20g or 0.25g/kg. TAR = percent time above range (>180mg/dL), TIR = percent time in recommended
glucose range (70-180mg/dL), and TBR = percent time below range (<70mg/dL) following bouts of
MVPA. * indicates statistical significance within sex category (p≤0.05).
In Aim 2 Results on Page 11:
“Following additional adjustment for tanner stage, significant interaction effects were also observed
between protein intake category and sex for TIR (p<0.01) and TAR(p<0.01) indicating that following a
high protein diet pattern may improve TIR, 16.3% (95% CI: 8.4%, 24.2%), and TAR, -16.9% (-25.3%, -
8.5%), to a greater extent among female adolescents. Figure 4 shows a comparison of the effects of a
high protein diet on post-exercise glycemia by insulin regimen, weight status, and sex.”
We added a Figure 4 on page 11 which provides an visual comparison of differences in the association
between daily protein intake category and post-exercise glycemia by insulin regimen, weight status and
sex:
Figure 4. Differences in the effect of consuming a higher protein diet (>1.2g/kg/day) on glycemia from the
end of MVPA bouts until the following morning by insulin regimen, weight status, and sex. CSII =
Continuous Subcutaneous Insulin Infusion, MDII = Multiple Daily Insulin Injections, OW/OB = Overweight
or Obesity, No OW/OB = no over-weight or obesity. TAR = percent time above range (>180mg/dL), TIR =
percent time in recommended glucose range (70-180mg/dL), TBR = percent time below range
(<70mg/dL) following bouts of MVPA.
We added the following text in the Discussion on page 13 where we are summarizing the main findings of
the study:
“Further, no significant interaction effects were observed between post-exercise protein intake and
exercise volume, insulin regimen, or weight status, however, a significant interaction effect was observed
for post-exercise protein intake and sex which indicated that increased post-exercise protein intake may
be associated with improved TIR and reduce TBR among female, but not male adolescents. Additionally,
daily protein intakes ≥ 1.2g/kg/day were associated with improved TIR and reduced TAR, but not TBR following MVPA bouts, with significant interaction effects observed indicating that female adolescents, those
with overweight/obesity, and those on multiple daily insulin injections may experience greater increases in
TIR and greater reductions in TAR with daily protein intakes >1.2 g/kg/day.”
“The findings of this study did not support the hypothesis that increasing post-exercise protein intake may
improve glycemia following exercise among adolescents with T1D overall, but it may be a beneficial
strategy among female adolescents with T1D.”
In the Discussion on Page 15, where we discuss potential explanations for the observed associations, we
made the following adjustments/additions:
“Additionally, elevated adiposity among individuals with overweight or obesity has been associated with
higher levels of insulin resistance in youth.52 In fact, among adolescents and adults with T1D, fat mass
has been shown to be positively related to post-exercise blood glucose and lean mass has been shown to
be inversely related to post-exercise blood glucose, indicating that body composition may play an
important role in the post-exercise glycemic response.53 Changes in insulin sensitivity may have
contributed to improved post-exercise glycemia to a greater extent among this population.
Differences in body composition between male and female participants may also help to explain the
differences in association we observed by sex as female participants in this study had higher estimated
bodyfat percentages compared to male participants (33.5% ± 5.9% vs 20.3% ± 4.2%). While we adjusted
our models for estimated body fat percentage, we may not fully account for other differences in body
composition, including differences in lean mass between male and female adolescents. Additionally,
previous studies in adolescents with T1D have shown that female adolescents oxidize more fat and less
carbohydrate during exercise compared to males.54 In populations without diabetes, this difference in
substrate utilization among women has been attributed in part to differing progesterone and estrogen
levels during the follicular versus luteal phase of the menstrual cycle and has been shown to be yield less
hepatic and muscular glycogen depletion during exercise among women.55 While speculative, these
differences in substrate utilization during exercise and the potential sparing of hepatic and muscle
glycogen may influence the post-exercise glycemic response among female adolescents. Studies are
needed, however, to elucidate the mechanisms by which protein affects glycemia and the influence of
factors such as insulin regimen, weight status and sex on this relationship in the unique metabolic context
of T1D.”
